# Development of a Rapid Epstein–Barr Virus Detection System Based on Recombinase Polymerase Amplification and a Lateral Flow Assay

**DOI:** 10.3390/v16010106

**Published:** 2024-01-11

**Authors:** Yidan Sun, Danni Tang, Nan Li, Yudong Wang, Meimei Yang, Chao Shen

**Affiliations:** 1College of Life Sciences, Wuhan University, Wuhan 430072, China; yidan97@whu.edu.cn; 2China Center for Type Culture Collection, Wuhan University, Wuhan 430072, China; danietang@163.com (D.T.); ln452602345@gmail.com (N.L.); wangliangsccsbx@163.com (Y.W.)

**Keywords:** RPA, LFA, EBV, cell quality control

## Abstract

The quality of cellular products used in biological research can directly impact the ability to obtain accurate results. Epstein–Barr virus (EBV) is a latent virus that spreads extensively worldwide, and cell lines used in experiments may carry EBV and pose an infection risk. The presence of EBV in a single cell line can contaminate other cell lines used in the same laboratory, affecting experimental results. We developed three EBV detection systems: (1) a polymerase chain reaction (PCR)-based detection system, (2) a recombinase polymerase amplification (RPA)-based detection system, and (3) a combined RPA-lateral flow assay (LFA) detection system. The minimum EBV detection limits were 1 × 10^3^ copy numbers for the RPA-based and RPA-LFA systems and 1 × 10^4^ copy numbers for the PCR-based system. Both the PCR and RPA detection systems were applied to 192 cell lines, and the results were consistent with those obtained by the EBV assay methods specified in the pharmaceutical industry standards of the People’s Republic of China. A total of 10 EBV-positive cell lines were identified. The combined RPA-LFA system is simple to operate, allowing for rapid result visualization. This system can be implemented in laboratories and cell banks as part of a daily quality control strategy to ensure cell quality and experimental safety and may represent a potential new technique for the rapid detection of EBV in clinical samples.

## 1. Introduction

Cell culture is currently a widely used biotechnology platform, and cell lines derived from various species and origins have become essential tools in the study of human metabolism and physiology [1]. Mammalian cell lines, such as Chinese hamster ovary cells, hybridoma cells, human embryonic kidney cells, and young hamster kidney cells, are commonly used for the development of antibodies and other drugs [2]. Additionally, insect-derived cell lines are becoming increasingly important in gene therapy research because they are useful for the production of recombinant proteins, viruses, and viral components [3]. As biological research continues to advance, the risks increase that cells will be misidentified or contaminated with other cell types or exogenous factors [4,5], including viruses, and such contamination can be difficult to detect and remediate. Although some viral infections result in morphological changes, such as cytopathic effects detectable by microscopy, other viral infections are associated with no visible changes in cellular appearance or alterations that occur slowly and are not easily observed. The viral contamination of biological cell cultures can be costly. For example, in 2009, Genzyme was required to pay USD 1.75 billion in fines due to viral contamination, in addition to reporting USD 1–3 billion in lost product sales [6]. Additionally, contamination impacts experimental results, potentially affecting experimental reproducibility and leading to wasted human and material resources. More importantly, viral contamination is a safety concern that poses a threat to human health. Vaccine products have been found to have been inadvertently contaminated with harmful viruses during production [7], and hemophiliacs treated with virus-infected plasma developed autoimmune deficiency virus infections [8], which eventually led to thousands of deaths in the 1980s and 1990s. Therefore, developing effective methods for detecting viral contamination in cells is essential.

Epstein–Barr virus (EBV) is a member of the human herpesvirus family, which includes eight viruses categorized into three subfamilies (α, β, and γ). EBV, also known as human herpesvirus type IV, belongs to the γ subfamily, genus *Lymphocryptovirus* [9]. EBV is a DNA virus with a genome size of approximately 170 kb that contains more than 100 open reading frames. EBV was first identified in 1964 in tissue samples from children suffering from African Hodgkin’s disease [10]. EBV is also recognized as a tumor-associated virus and was declared a class I carcinogen by the International Agency for Research on Cancer and the World Health Organization in the late 1990s [11]. EBV is transmitted primarily through direct contact with saliva, although aerosol-mediated transmission also occurs, and the virus infects lymphocytes and oropharyngeal epithelial cells. Saliva from first-time infected individuals present with very high levels of EBV DNA, which can persist for several months. Although EBV DNA levels subside over time following new infections, EBV can be periodically released into the oral secretions of carriers [12]. During the early stages of infection, the virus proliferates in the lymphocytes of the pharynx, after which the virus enters the bloodstream and spreads to the lymphatic system. EBV displays prolonged latency in lymphocytes, interfering with immune functions and potentially inducing cell proliferation and transformation. EBV infection involves many organ systems and is often misdiagnosed or underdiagnosed. Therefore, early diagnosis and rational treatment are extremely important.

Cell lines that contain EBV genes can be broadly classified into two categories. One category includes cell lines that are transformed by latent EBV infection, such as the in vitro transformation of resting B cells into immortalized lymphoblastoid cell lines. In 1973, a study reported that B95-8 cells transformed from marmoset blood leukocytes regularly released high EBV titers and displayed transforming activity [13]. The second category includes cell lines contaminated with EBV. During latent EBV infection, the virus is detectable in the nucleus in a ring form, linked to the chromatin of the host genome by the Epstein–Barr nuclear antigen 1 (EBNA1) protein [14]; genes expressed during latent EBV infections are referred to as latent genes. Multiple reports in the literature have described the detection of EBV infection in various cell types maintained in cell culture banks [15,16]; these infections can be attributed to the presence of an existing EBV infection during the initial process of cell line establishment, EBV contamination of culturing materials, or improper manipulation by the experimental staff.

Existing tests to detect EBV can be divided into three categories: nucleic acid assays, serological assays, and in situ hybridization assays. Nucleic acid assays include gene-specific amplification techniques and whole-genome sequencing. Gene-specific amplification techniques include polymerase chain reaction (PCR) and isothermal amplification. Currently the most widely used assay, PCR is a mature and reliable molecular method, and quantitative PCR (qPCR) can be used to monitor disease progression or treatment efficacy by detecting changes in the EBV load in blood [17]. Isothermal amplification techniques, such as recombinant enzyme-mediated isothermal amplification and loop-mediated isothermal amplification (LAMP), are easy to perform, have limited equipment requirements, return rapid responses, and display high sensitivity. Wang et al. utilized a recombinase-aided amplification (RAA) method to detect EBV in whole blood specimens and serum [18]. Iwata et al. used the LAMP technique to detect EBV in serum and pharyngeal swabs [19]; however, designing primers for use in the LAMP technique is complicated and can be difficult to replicate. Although whole-genome sequencing can be used to accurately evaluate the full EBV sequence [20], this method is time-consuming, expensive, and not conducive to high-volume clinical screening. Serological assays are based on the detection of relevant antibodies produced in response to EBV infection. Commonly used serological assays utilize techniques such as immunofluorescence staining, enzyme immunoassays, chemiluminescence immunoassays, Western blotting, and immunofluorescence reactions [21]. In situ hybridization assays to detect Epstein–Barr early RNA(EBER), a small EBV-encoded RNA that is continuously transcribed and expressed after infection, are the current gold standard for detecting EBV infection in clinic. However, this technique can only be applied to tissue samples and is generally limited to the clinical diagnosis of EBV-related diseases, such as cancer and lymphoma. Therefore, a simple system that allows for the rapid detection of EBV in multiple contexts, including both cell culture and tissue samples, remains necessary.

Recombinase polymerase amplification (RPA) is a highly sensitive and selective isothermal amplification technique that can be performed at 37–42 °C and can be used to amplify a large number of samples in a short period of time. RPA can be performed on small sample volumes and can be applied to a variety of sample types, including microorganisms, body fluids, surgical biopsies, organ tissues, and plant and animal products [22]. RPA, first introduced in 2006 by Niall Armes of ASM Scientific Ltd., Cambridge, UK [23], relies on modified homologous recombination mechanisms. In addition to reaction cofactors, such as DNA polymerase and energy-generating molecules, the standard RPA reaction reagent contains three key proteins: T4 uvsX recombinase, T4 uvsY recombinase loading factor, and T4 gp32 single-strand binding protein [23]. RPA reagents are commercially available, and the basic RPA reaction kit can be augmented by additional commercially available kits that use different probes: exo, fpg, and nfo [24]. The exo and fpg probes are typically used for real-time detection, whereas the nfo probe is typically used for detection systems that use lateral flow dipsticks. RPA technology is well suited for on-site detection in low-resource environments and represents a promising platform for the development of amplification-based detection systems.

In this study, we combined RPA technology with a lateral flow assay (LFA) to develop an RPA-LFA detection system for application to the rapid and bulk screening of EBV contamination in cell lines stored by cell banks and as a means to conduct regular and daily inspection of cell lines used in biological experiments, ensuring the quality of cell lines and the safety of experimental personnel. This system also demonstrates high potential for clinical adaptation to improve EBV detection in blood and tissue samples.

## 2. Materials and Methods

### 2.1. Cells

The cells used in this study were provided by the China Center for Type Culture Collection (CCTCC, Wuhan, China), and the list of cells is shown in Appendix A.

The presence of viral gene sequences in some of these cell lines has long been recognized [25,26,27,28], including human papillomavirus type 18 in HeLa cells, human papillomavirus type 16 in SiHa cells, hepatitis B virus in Hep-G2/2.2.15 cells, and bovine viral diarrhea virus in RK13 cells.

### 2.2. Nucleic Acid Extraction

#### 2.2.1. Cell DNA Extraction

Cells cultured in T25 culture flasks that exhibited favorable growth conditions were digested, centrifuged, and suspended, and DNA was extracted using a blood/cell/tissue genomic DNA extraction kit from Tiangen Biotech (Beijing, China) Co., Ltd., according to the manufacturer’s instructions. The cellular DNA concentration was measured using a NanoDrop 2000 (Thermo Fisher, Waltham, MA, USA) and then diluted to a working concentration (50 ng/μL) using ddH_2_O.

#### 2.2.2. EBV DNA Extraction

B95-8 cells were routinely cultured in a CO_2_ incubator, and viral DNA was extracted 1 week after the cells became nearly confluent. Culture flasks were freeze–thawed three times at −80 °C and 37 °C, and the culture fluid was obtained and centrifuged at 12,000 rpm for 10 min. Then, the supernatant was filtered through a 0.45 μm sterile filter to obtain EBV-containing fluid, which was stored at −80 °C. The virus solution was processed using an OMEGA Viral DNA kit (Omega Bio-tek, Norcross, GA, USA) to obtain EBV DNA, and the DNA quantity and quality were measured using a NanoDrop 2000 (Thermo Fisher Scientific Inc., Waltham, MA, USA).

### 2.3. Construction of Standard EBV Plasmids

The BZLF-1 gene fragment (GenBank: MK_540470) of EBV was synthesized using primers (Forward: 5′-CCTGGTCATCCTTTGCCA-3′; Reverse: 5′-TGCTTCGTTATAGCCGTAGT-3′) and inserted into the pMD18-T vector (TaKaRa D1010A, Takara Bio, Dalian, China). Then, the gene sequence accuracy was verified by sequencing (Appendix A). The recombinant plasmids were extracted using a TIAN Pure Midi Plasmid Kit from TIANGEN BIOTECH (Beijing, China) Co., Ltd., and the plasmid concentration was measured using a NanoDrop microvolume spectrophotometer from Thermo Fisher Scientific Inc. (Waltham, MA, USA). The DNA copy number of the recombinant EBV plasmid was calculated using the following equation:DNA copy number (copies/µL) = concentration (ng/µL) × 10^−9^ × 6.022 × 10^23^ (copies/mol)/[clone size (bp) × 660 (g/mol/bp)] 

Then, the DNA was serially diluted 10-fold from 1.0 × 10^10^ to 1.0 × 10^0^ copies/µL.

### 2.4. Measurement of Viral DNA Concentration

The previously extracted viral DNA and the gradient of 10 diluted plasmids were used as templates, and 1 μL of each was added to a 96-well plate, with five replicates of each group and several negative controls. Primers (1 μL) (Forward: 5′-CCTGGTCATCCTTTGCCA-3′; Reverse: 5′-TGCTTCGTTATAGCCGTAGT-3′), 10 μL of SYBR Green fluorescent dye, and 7 μL of ddH_2_O were added to each well, and the wells were assayed using a CFX96 Touch Real-Time PCR Detection System from Bio-Rad Laboratories (Hercules, CA, USA). The heating program included a preheating step for 5 min at 95 °C and 40 cycles of 95 °C for 30 s, 58 °C for 30 s, and 72 °C for 30 s. The melting curve was measured from 65 to 95 °C in 0.5 °C intervals. At least six standard concentrations were retained from the measured data, and a standard curve was generated using the cycle threshold (CT) values of plasmids with different concentrations. The Pearson coefficient of the standard curve was above 0.99, and the CT value of the sample was used to calculate the virus concentration.

### 2.5. PCR for EBV Detection

#### 2.5.1. Primer Screening

Five pairs of primers were designed for the EBV genome (GenBank: V01555) using DNAMAN (version 8.0) primer design software (Table 1), with amplicon sizes of 265, 231, 134, 159, and 95 bp. The amplification system contained 10 μL of Premix-Taq, 0.5 μL of forward primer (10 μM), 0.5 μL of reverse primer (10 μM), and ddH_2_O, which was added to supplement the 20 μL system. The PCR reaction conditions included an initial denaturation temperature of 95 °C for 3 min; 35 cycles of denaturation at 95 °C for 15 s, annealing at 56 °C for 15 s, and extension at 72 °C for 60 s; and a final extension at 72 °C for 5 min. The resulting PCR products were subjected to 2% agarose gel electrophoresis.

#### 2.5.2. Determination of the Test Sample Type

Different types of templates (cellular supernatant, cellular precipitate, and extracted cellular DNA) were subjected to PCR amplification under the conditions described in Section 2.5.1. Following cell digestion and centrifugation, the cellular supernatant was removed, and an equal volume of cell lysis buffer (1% Triton) was added to the resultant precipitate to obtain the cellular precipitate. The other components and the total amount of the reaction system were identical, and the template and ddH_2_O volumes were slightly adjusted, including 5 μL for the cell supernatant template, 2 μL for the cell precipitate template, and 1 μL for the cell extract DNA template (50 ng/μL). The cell supernatant was obtained from the supernatant of well grown cells after digestion and centrifugation, the cell precipitate consisted of the precipitated portion, and the extracted cellular DNA was obtained using a blood/cell/tissue genomic DNA extraction kit from Tiangen Biotech (Beijing, China) Co., Ltd.

#### 2.5.3. Sensitivity Evaluation

EBV DNA was diluted with ddH_2_O to produce a concentration gradient of 1 × 10^6^, 1 × 10^5^, 1 × 10^4^, 1 × 10^3^, 1 × 10^2^, 1 × 10^1^, and 1 × 10^0^ copies/μL for use as templates, and 1 μL was used for PCR amplification.

### 2.6. RPA Method of EBV Detection

#### 2.6.1. Basic RPA System

A Twist Amp Basic kit was used with the primers from the PCR method, and 50 ng/μL of cellular DNA was used as a template. Then, 29.5 μL of buffer, 2 μL of unlabeled upstream and downstream primers (10 μM each), 1 μL of extracted cellular DNA, and 13 μL of ddH_2_O were added to the reaction tube, and finally, 2.5 μL of 280 mmol/L magnesium acetate solution was added to start the reaction. The reaction solution was mixed well and placed in a thermal incubator for 5–20 min at 39 °C.

#### 2.6.2. Optimization of the Reaction Conditions

B95-8 cellular DNA (50 ng/μL) was used as a template, and the reaction time and temperature were optimized. The reaction time was set to 5, 10, 15, or 20 min to observe amplification with a constant reaction temperature of 39 °C. In an additional assessment, the reaction time was fixed at 15 min for the initial reaction temperatures tested, including 25 °C, 35 °C, and 45 °C. Later, the process was refined in 35 °C, 37 °C, 39 °C, 41 °C, and 43 °C to observe the amplification of RPA reactions at different temperatures.

#### 2.6.3. Sensitivity Evaluation

The DNA was diluted with ddH_2_O to produce a gradient of 1 × 10^6^, 1 × 10^5^, 1 × 10^4^, 1 × 10^3^, 1 × 10^2^, 1 × 10^1^, and 1 × 10^0^ copies/μL for use as templates, and 1 μL was used for RPA amplification as described in Section 2.6.2 and reacted at 39 °C for 15 min.

### 2.7. RPA-LFA Method of EBV Detection

#### 2.7.1. Basic RPA-LFA System

One-step method: The primers described in Table 1 were labeled, i.e., the forward primer was labeled with FITC at the 5′ end, and the reverse primer was labeled with biotin at the 5′ end. The remaining reagents were the same as those used in the basic RPA system, and the reaction was performed at 39 °C for 20 min. The amplification product was diluted 1:100 with diluent, and 50 μL was added dropwise to the spotting area of the lateral flow dipstick. The results were observed after 5 min.

Two-step method: Step 1: Biotin labeling of the reverse primer only. The remaining reagents were the same as those used in the basic RPA system, and the reaction was performed at 39 °C for 20 min. Step 2: The FITC-labeled probe (0.6 μL), 1.5 μL of nfo, and 5 μL of nfo buffer were added to the reaction product of step 1, and the reaction was performed at 39 °C for 20 min. (the nfo and nfo buffer are from FastDigest, Inc., Waltham, MA, USA) The final amplification product was diluted with diluent at a ratio of 1:100, and then 50 μL was added dropwise to the lateral flow dipstick. The results were observed after 5 min. The sequences of the probes and labeled primers are listed below (Table 2).

A customized lateral flow dipstick was purchased from Wuhan Aoke Botai Biotechnology Co. (Wuhan, China).

#### 2.7.2. Optimization of Reaction Conditions

B95-8 cellular DNA (50 ng/μL) was used as the reaction template. Optimized reaction temperatures and times were selected for each step of the two-step reaction.

Reaction time: Step 1 of the two-step method was adjusted. The reaction time of step 1 of the common RPA reaction was set to 5, 10, 15, or 20 min, and step 2 was fixed at 15 min. Then, the reaction time of step 2 of the two-step method was optimized. The time of step 1 was fixed to 5 min, and step 2 of the enzymatic amplification was set to 5, 10, or 15 min.

Reaction temperature: The reaction time was fixed at 15 min for steps 1 and 2, and the sample reaction temperatures were set to 35 °C, 37 °C, 39 °C, 41 °C, and 43 °C for steps 1 and 2.

#### 2.7.3. Sensitivity Evaluation

The two-step amplification method was used. The EBV plasmid was diluted to a gradient of 1 × 10^6^, 1 × 10^5^, 1 × 10^4^, 1 × 10^3^, 1 × 10^2^, 1 × 10^1^, and 1 × 10^0^ copies/μL, and 1 μL of each concentration was used as a template for amplification. The reaction temperature was set to 39 °C. The amplification time was set to 15 min for both steps.

### 2.8. The Detection of Cell Lines

To evaluate the applicability of the new developed method (RPA-LFA), 192 cell lines were tested in parallel using commercial quantitative real-time polymerase chain reaction (qPCR, DaAn Gene, Guangzhou, China), traditional PCR, and RPA.

## 3. Results

### 3.1. Screening of Amplification Primers and Sample Types

To optimize the subsequent reactions, we first screened the amplification primers using PCR amplification. Five pairs of primers were designed, #1, #2, #3, #4, and #5, and 1 μL of B95-8 cell template DNA (50 ng/μL) was added to each group for PCR amplification; ddH_2_O was used as a blank control. The amplified samples were examined by agarose gel electrophoresis, and the amplification product sizes corresponding to the five primer pairs were 265, 231, 134, 159, and 95 bp, respectively. All sizes were in accordance with the expected results (Figure 1A), and the sequencing results all indicated that the primers were EBV sequences. Among them, primer pairs #1, #2, and #3 had clear and uniform target bands, while the amplification bands of primer pairs #4 and #5 were relatively weak. Therefore, the first three primer pairs could be used for the PCR amplification of EBV. Primer pair #1 was deemed the most suitable for use in subsequent experiments because this primer set produced a moderately sized product that resulted in obvious bands. The EBV fragment detected by primer pairs #1 is shown in Figure 1C.

The type of sample selected for amplification is also important for EBV detection. To ensure the accuracy of results and convenience of the assay, PCR amplification was first attempted using three sample types: cell supernatant (obtained by direct aspiration of cells after centrifugation), cell precipitate, and cellular DNA. HeLa (EBV-negative), B95-8 (EBV-positive), and Raji (EBV-positive) cells were used to verify the accuracy of the amplification results (Figure 1B). When cellular DNA was used as the assay sample, both the B95-8 and Raji samples showed obvious target bands, while HeLa samples showed no target bands, consistent with the expected results (see Figure 1B). When the cell supernatant was selected as the assay sample, none of the positive cell samples produced a clear band. When cell precipitate was used, unclear target bands were observed for B95-8 cells, and no obvious bands appeared in the Raji cell sample, indicating that the assay was not accurate and cell precipitate could not be used directly as a template for the assay. We repeated the above experiments three times, and the results were consistent. These experimental results indicate that EBV detection requires extracted cellular DNA as a template to ensure accurate results.

### 3.2. Establishment of a Reaction System for EBV Detection by the RPA-LFA System

To more accurately compare the difference between PCR analysis and the RPA-LFA method in detecting EBV, we used the same sample types for RPA-LFA amplification.

First, the experimental results of the RPA method were validated using the same primers and sample types used for the PCR method. We used B95-8 cellular DNA as the positive control, HeLa cell DNA as the negative control, and ddH_2_O as the blank control for amplification. The RPA amplification products were subjected to agarose gel electrophoresis. Clear bands were observed for the positive control, and no bands were observed for the negative and blank controls (Figure 2A), indicating that the primers and sample types are suitable for the RPA system.

Second, an RPA-LFA visualization system for EBV detection was developed. We performed the RPA reaction using a forward primer with a FITC tag and a reverse primer with a biotin tag to generate isothermal amplification products with both tags. After the RPA reaction was complete, the product was diluted and added dropwise to a lateral flow dipstick to observe the assay results (Figure 2B). Although the positive samples showed positive results, weak positive bands were observed in both the blank control and negative control. We sequenced these two sets of suspicious positive products and found no EBV fragments. Therefore, we excluded the possibility of experimental contamination. Based on these results, we concluded that the primer dimer in the reaction system produced false-positive bands on the test strips; therefore, a probe was introduced. Initially, we performed the RPA reaction using unlabeled forward primers, biotin-labeled reverse primers, and a FITC-labeled probe (containing tetrahydrofuran (THF)). Because the probe contained a THF internal dibasic site that needed to be cleaved by nucleic acid endonuclease IV (also known as nfo) to expose the 3′-OH group for extension, we also added nfo and an nfo companion buffer to the reaction system. However, this reaction system did not amplify the target bands (Figure 2C). Therefore, we adjusted the system composition again. To determine which of the newly added components affected the normal reaction, different templates (ddH_2_O, HeLa DNA, and B95-8 DNA) were chosen to validate the initial RPA system. Meanwhile, the RPA reaction system lacking a B95-8 DNA template was supplemented with different components (probe, nfo, and nfo buffer) in turn. The target bands were weakly amplified when the amplification system contained nfo buffer, but the target fragments could be amplified under other conditions (Figure 2D). Therefore, we removed the nfo buffer from the reaction system, kept only the original RPA system, and added nfo and a probe to observe the results. Although the positive sample showed the amplification of the target fragment and the negative and blank controls had no target bands, a blank control amplification product was still detected (Figure 2E,F).

Eventually, we split the above amplification step into two steps. In the first step, the samples were amplified by conventional RPA isothermal amplification (forward primer without a label and reverse primer with a biotin label) and reacted at 39 °C for 15 min. In the second step, the probe, nfo, and nfo buffer were added to the amplification product from the previous step, and the reaction was carried out at 39 °C for 15 min. The reaction was visualized with a lateral flow dipstick after completion. After this improvement of the reaction system, the false-positive phenomenon disappeared in the negative and blank groups (Figure 2G,H). Thus, the introduction of the probe solved the false-positive phenomenon, and the RPA-LFA detection system for EBV was completed (all RPA-LFA detection systems described hereafter in this paper are two-step systems).

### 3.3. Determination of the RPA-LFA Reaction Temperature

To further optimize the RPA procedure, the RPA reaction temperature was examined. First, the reaction temperature was set to 25° C, 35 °C, or 45 °C, and the reaction time was set to 20 min. Three cell types, HeLa, B95-8, and Raji cells, and the positive control (target fragment, 143 bp) provided in the kit were used for verification. No band appeared in the positive sample when the reaction temperature was 25 °C, but the target fragment was amplified at 35 °C and 45 °C (Figure 3A). To further determine the optimum temperature, we used only EBV-positive B95-8 DNA amplified at 35 °C, 37 °C, 39 °C, 41 °C, and 43 °C. The results suggested that the RPA system could amplify the target bands under all conditions, but the bands were brightest at 39 °C (Figure 3B). The optimum temperature was then determined for the RPA-LFA system (two-step method), and the results were similar to those of the RPA system (Figure 3C). Therefore, 39 °C was selected as the optimum reaction temperature for this system.

### 3.4. Determination of the RPA-LFA Reaction Time

After the reaction temperature was optimized at 39 °C, the reaction time was further adjusted to minimize the experimental time. We used B95-8 cell DNA as the amplification template and performed the RPA reaction for 5, 10, 15, or 20 min. The reaction products were examined by agarose gel electrophoresis immediately after the reaction. No clear bands could be observed for the 5 min reaction, but specific bands gradually became obvious as the reaction time was extended. When the reaction time was 15 min, the bands were sufficiently clear, and at 20 min, the bands were not notably different from those obtained in the 15 min reaction (Figure 4A). Therefore, 15 min was selected as the optimal reaction time.

The reaction times of both steps for the RPA-LFA system were carefully analyzed. First, the reaction time of the second step was maintained at 15 min, and the reaction time of the first step was set to 5, 10, 15, or 20 min to investigate the effect of the first enzymatic amplification step on amplification. Agarose gel electrophoresis showed positive target fragments in the reactions, with the first step ranging from 5 to 20 min and up to 15 min for the second set, and the band clarity and brightness were proportional to the reaction time (Figure 4B). These results show that positive fragments can be detected when the first reaction step is 5 min. Therefore, in subsequent experiments, the reaction time of the first step was set to 5 min, and the reaction time of the second step was set to 5, 10, or 15 min to verify the amplification effect. Agarose gel electrophoresis showed that the target bands were amplified only when the first step was isothermal for 5 min and the reaction time of the second step was 5–15 min (Figure 4C). The reaction time was verified by combining the RPA product with a lateral flow dipstick, and the results were consistent with those described above (Figure 4D,E).

According to the reaction results, the RPA-LFA method was more convenient and rapid than expected, and obvious bands could be detected when both steps were set to 5 min. As the reaction time increased, the results became clearer. To ensure the accuracy of the subsequent experimental results, we used 15 min for both steps.

### 3.5. Comparison of the Sensitivity of the PCR and RPA-LFA Methods for EBV Detection

We extracted pure EBV DNA from B95-8 cells and calculated its concentration using the constructed EBV nuclear gene plasmids. Additionally, the concentration of the constructed EBV nuclear plasmid was measured, and the sample was diluted with ddH_2_O to produce a gradient of 10^0^ to 10^10^ copies/μL. The two solutions were subjected to RT-qPCR to obtain a standard curve for the EBV nuclear genes using the LOG gene copy number as the horizontal coordinate and the CT value as the vertical coordinate. The CT value of 10^0^ copies/μL was discarded because the deviation of this point was large, and a standard curve (Figure 5A) was plotted. The CT value was inversely proportional to the gene copy number, and the R^2^ value of the curve was 0.99. Based on this standard curve, the concentration of EBV DNA extracted from B95-8 cells was calculated to be 1.31 × 10^6^ copies/μL, and the DNA was diluted with ddH_2_O to 1 × 10^6^ copies/μL. The EBV DNA was further diluted with ddH_2_O to produce a gradient of 1 × 10^6^, 1 × 10^5^, 1 × 10^4^, 1 × 10^3^, 1 × 10^2^, 1 × 10^1^, and 1 × 10^0^ copies/μL.

First, 1 μL from each of the diluted gradient solutions was used as a PCR template and was diluted to 1 × 10^4^ copies/μL, but when the viral solution was diluted to 1 × 10^3^ copies/μL or less, no clear bands could be observed upon electrophoresis (Figure 5B). Therefore, the lower limit of the virus copy number for the PCR detection system was determined as 1 × 10^4^. Then, we tested the sensitivity of the RPA system. We used a reaction temperature of 39 °C and a reaction time of 15 min. Agarose gel electrophoresis showed visible target bands at EBV concentrations greater than or equal to 1 × 10^3^ copies/μL, but specific bands could not be observed at concentrations of 1 × 10^2^ copies/μL or less (Figure 5B). In particular, when the RPA product was applied to a lateral flow dipstick, the optimal reaction time of both steps was determined to be 15 min, and the optimal reaction temperature was 39 °C. No positive bands were detected by the RPA-LFA system when the virus copy number in the amplification system was less than 1 × 10^3^ (Figure 5C), which was consistent with the RPA results.

The agarose gel electrophoresis results indicated that both the RPA and RPA-LFA systems were more sensitive than the PCR system, and the RPA-LFA system could visualize the results quickly while ensuring high sensitivity.

### 3.6. Use of the PCR and RPA Systems to Detect EBV in Different Cell Lines

After establishing the PCR and RPA assay systems and determining the sensitivity of the assay, we used these two systems to detect EBV in 192 cell lines from the CCTCC. The PCR method identified 11 cell lines, B95-8, Raji, NK-92, Daudi, ARH-77, JVM-2, Farage, MC/CAR, CCRF-SB, A-431, and A-204 cells, that were positive for EBV, while the remaining cell lines were negative. Additionally, the RPA results showed that a total of 10 cell lines, B95-8, Raji, NK-92, Daudi, ARH-77, JVM-2, Farage, MC/CAR, CCRF-SB, and A-431 cells, were positive for EBV, and the remaining cell lines were negative (Appendix A). One cell line, A204, showed a discrepancy in the assay results. Therefore, we performed PCR and RPA analyses on this cell line again after re-extracting the same batch of A204 cells used for the previous assay. Both assays were negative (Appendix A). Therefore, it was speculated that the first positive result could be due to an operational error, and the cells were finally determined to be EBV-negative. A total of 10 EBV-positive cell lines were detected by the two assays, which showed consistent results. All cell lines tested and graphs of the agarose gel electrophoresis results for both methods are shown in the Appendix A.

In order to further confirm the reliability of the results of the two methods, we introduced an EBV nucleic acid detection kit (fluorescent PCR) prescribed by the Pharmaceutical Industry Standard of the People’s Republic of China. After testing 192 cell lines with the kit, the assay presented 10 positive cell lines, consistent with PCR and RPA results.

### 3.7. RPA-LFA Specificity Evaluation

To verify the specificity of the RPA-LFA system, we screened several cell lines containing endogenous viruses among the 192 tested cell lines: HeLa (containing human papillomavirus type 18), SiHa (containing human papillomavirus type 16), Hep-G2/2.2.15 (containing hepatitis B virus), and RK13 (containing bovine viral diarrhea virus). The RPA-LFA method was validated using these cell lines, and the results were negative for EBV (Figure 6A), indicating that the system has good specificity. In addition, we randomly selected cell lines for further validation, including three EBV-negative cell lines (A-204, H9, and HTR-8) and the remaining nine cell lines (NK-92, Daudi, ARH-77, Raji, JVM-2, A-431, Farage, MC/CAR, and CCRF-SB) that showed EBV positivity in previous tests. All cell detection results were consistent with expectations (Figure 6B,C), and the RPA-LFA detection results were consistent with those of the PCR and RPA methods.

In summary, the RPA-LFA system not only allows for the rapid visualization of results but also has the same accuracy as the RPA system.

## 4. Discussion

Cell culture is the most fundamental step in biological experiments, but human error and contamination carried by the cells themselves can have a dramatic effect on experimental research and the cellular products produced. Several cell lines have been shown to be persistently infected by EBV; for example, EBV can infect B lymphocytes and convert them into a continuously proliferating lymphoblastoid-like cell line [29]. Additionally, EBV, a herpesvirus that widely circulates in society, may also be introduced into cells by operators who carry EBV themselves due to inadvertent manipulation, resulting in cellular virus contamination. Existing PCR-based methods for EBV detection have drawbacks, such as long detection periods, complicated operations, and limited applications. Therefore, we used an RPA method combined with an LFA to establish a rapid EBV detection system. The operating procedure of the system is shown in Figure 7. The RPA-LFA system consists of three parts: sample preparation, nucleic acid amplification, and use of LFA test strips to visualize the results. Cellular DNA is first extracted as the sample type for the assay and then amplified by RPA for 5–15 min at 39 °C. The nfo probe is introduced during amplification to bind to the test strip for the rapid visualization of the results.

The addition of probes to the system is crucial. Initially, we only replaced the normal forward and reverse primers during RPA amplification with FITC-labeled forward primers and biotin-labeled reverse primers, but the negative and blank controls consistently showed false positives when applied to the lateral flow dipstick. By varying the primer content, we observed that the false-positive bands were proportional to the primer content within a certain range. Therefore, the primers were determined to be the main factor causing this phenomenon. The initial speculation was that some of the forward and reverse primers formed primer dimers, and such primer dimers with two markers at the same time were eventually titrated on the test strip and captured to form positive bands. The same false-positive phenomenon was also observed in a study of avian influenza virus detection using nfo probes [30], in which Wang et al. introduced probes and base substitutions to eliminate the false-positive phenomenon. Therefore, we introduced a labeled probe in addition to the normal RPA assay, and only the reverse primer was labeled. Nfo was added to cut off the C3-Spacer of the 3′ end of the probe to obtain a new target fragment with both labels and eliminate the interference of the primer dimer. Most studies introducing nfo probes for RPA have used the TwistAmp^®^ nfo kit from TwistDx (Maidenhead, UK). For example, Greeshma et al. used this kit for the rapid detection of black pepper infestation by pepper yellow mottle virus [31], and Velasco et al. used the kit for quality control of seafood [32]. However, the TwistAmp^®^ nfo kit is expensive and not easy to purchase, and the long procurement period did not meet the requirements of the assay. To reduce the assay cost and optimize the procedure, we built our own amplification system containing nfo. At first, we added the designed probe, nfo, and its buffer directly to the RPA amplification system according to the recommendation in the instruction manual, but little amplification was observed on the agarose gel after adding these reagents. We then created a two-step method for EBV detection, i.e., amplification with labeled reverse primer according to the normal RPA system, followed by re-amplification with the addition of the probe, nfo, and its buffer, which effectively eliminated false positives. We also determined the minimum reaction time and found that a positive test could still be detected when the total time of the two-step reaction was shortened to 10 min.

Another interesting finding of this study is that nine of the 10 EBV-positive cell lines have been associated with EBV in the literature, but an association has not been reported for A-431. For example, Uphoff et al. screened cells for EBV, and their positive test results included six of the EBV-positive cell lines in this study: B95-8, ARH-77, Daudi, JVM-2, NK-92, and Raji [15]. Farage, CCRF-SB, and MC/CAR are listed as EBV transformants in the NCBI database. Nevertheless, no mention of A-431 cells containing EBV fragments within their genes has been reported in the published literature to date. Subsequent examinations of other A-431 cell batches showed that only one batch was positive for EBV according to both the PCR and RPA methods; the remaining batches were negative (Appendix A). We hypothesized that the previous batch of A-431 cells may have been contaminated by the external environment during late culture, resulting in EBV positivity, or that the human-derived cells may have been infected with EBV when the cell line was initially established. In either case, the EBV contamination of cells presents an obstacle to the health of the operator and the conduct of biological experiments, further suggesting that cellular EBV testing is imperative.

Although EBV detection methods in the field using isothermal amplification technology exist, the detection system established in this study still has certain advantages. The use of LAMP for the detection of EBV in serum and pharyngeal swabs [19] requires a complex primer design, whereas RPA primer design is simple and easy to perform. The isothermal amplification of nucleic acids using RAA has been applied in a manner similar to RPA, including in a study by Yuan Gao et al. [18] to detect the sensitivity and specificity of RAA for detecting EBV in extracted nucleic acids. Additionally, Jing-yi Li et al. used the RAA method in combination with magnetic beads enriched with recombinant human mannan-binding lectin to detect low-carbon-load EBV in blood [33]. However, the RPA method was developed first and is more mature and stable than the RAA method, which does have some advantages. Furthermore, the RPA-LFA detection system that we established has only been shown to be feasible at the cellular quality control level, and further applications for clinical testing can be explored using different types of clinical samples, such as blood and pharyngeal swabs. Moreover, more possibilities for rapid the binding of target fragments in this amplification system can be explored to reduce the time required to extract DNA and simplify the overall process. For example, microfluidic biochips based on microelectromechanical systems technology have been reported to enable rapid cell lysis, resulting in the acquisition of amplifiable genomic DNA in less than 20 min [34]. Consequently, the establishment of the RPA-LFA rapid EBV detection system has great significance in the pharmaceutical field, including for cell quality control and clinical testing.

The RPA-LFA system requires further improvement. First, the two-step method established in this study has the potential to be integrated into a one-step amplification method. TwistDx published a study on RPA technology in 2018 [35] that contains detailed kit components and can provide a reference for subsequent experimental component adjustment so that the RPA amplification phase and nfo probe digestion phase do not interfere with each other and react simultaneously. Second, the use of different types of reaction equipment can be explored to reduce the assay cost. Instead of using a dedicated instrument to maintain the isothermal amplification of reactions, other more cost-effective instruments, such as constant temperature metal baths and boiling water baths, can be used, and human-generated heat has been demonstrated for the device-free amplification of HIV [36]. In addition, the introduction of other types of probes can also enrich the detection capabilities of the reaction system. Currently, only the qualitative and rough quantitative detection of EBV is possible, but the lowest detection limit of 16 copies of the monkeypox virus was recently reported by introducing exo probes [37]. Therefore, the introduction of other types of probes and the addition of common qPCR for more accurate quantitative detection can be considered in the future. Finally, the sensitivity of the RPA-LFA system can be improved. In this study, a single reaction system (50 μL) could detect 1 × 10^3^ copies of EBV, but in a similar study of RPA combined with an LFA for virus detection, the detection sensitivity was as low as 200 copies [38]. We previously demonstrated that changing the probe amount does not change the reaction sensitivity; therefore, other conditions need to be explored to improve the detection sensitivity of the RPA-LFA system. When customizing the lateral flow strips, AuNPs with larger particle sizes were selected as labeling materials to improve sensitivity. Choi et al. used two different sizes of AuNPs to achieve the sensitive detection of cardiac troponin I (cTnI) by LFA, and the difference in sensitivity between different sizes of AuNPs was obvious [39]. On the other hand, combinations with other methods can also be considered, such as those used in a study by Jonathan et al., in which gold nanoparticles were used as amplicons in combination with electrochemical methods, presenting higher detection sensitivity through specific binding of amplicons [40].

The RPA-LFA rapid EBV detection system established in this study is simple to operate, does not require expensive equipment, and allows for rapid visualization, which is important for the quality management of cell banks. It also provides a new method for EBV detection in clinical samples and other applications.

## Figures and Tables

**Figure 1 viruses-16-00106-f001:**
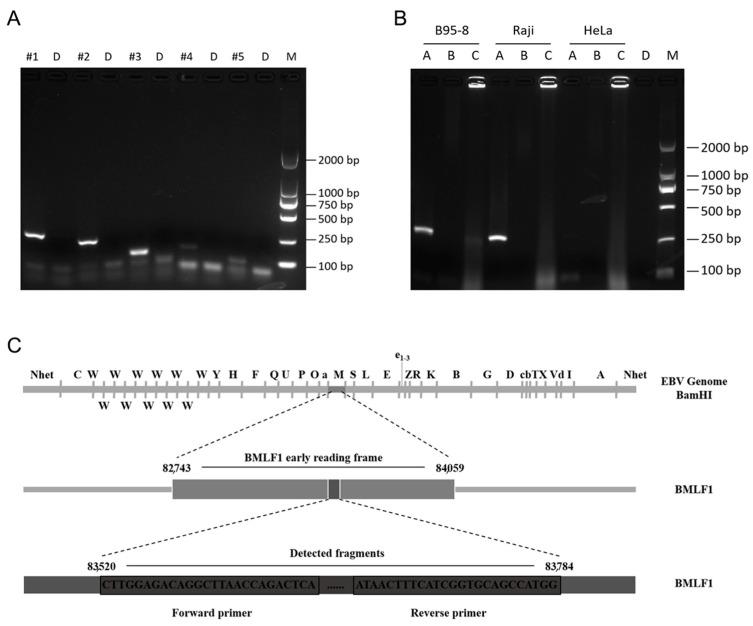
Primer screening and sample type selection. (**A**) Agarose gel electrophoresis results after PCR amplification with different primers. #1, #2, #3, #4, #5: five different primer pairs with target fragment sizes of 265, 231, 134, 159, and 95 bp, respectively; D: ddH_2_O; M: DL2000 marker; (**B**) Agarose gel electrophoresis results after PCR amplification of different sample types. A: cellular DNA; B: cell supernatant; C: cell precipitate; D: ddH_2_O; M: DL2000 marker. (**C**) Location of the EBV fragment detected by primer pairs #1 on the BamHI restriction map of the prototype EBV B95-8 genome.

**Figure 2 viruses-16-00106-f002:**
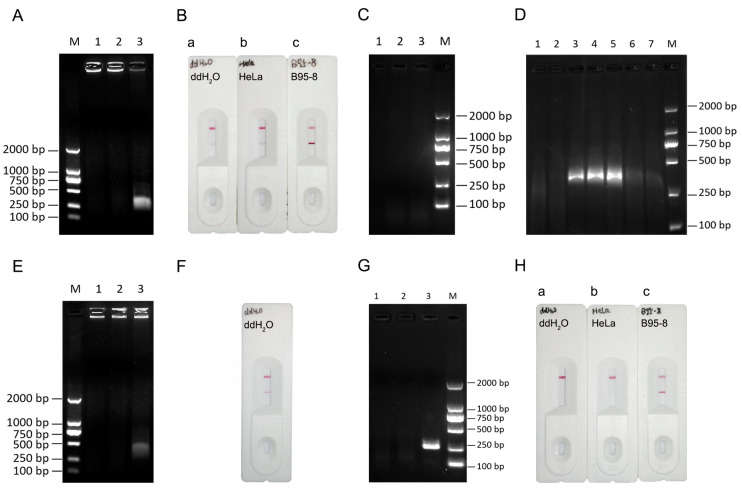
Establishment of the RPA-LFA detection system for EBV. (**A**) Validation of the RPA system. 1: ddH2O; 2: HeLa cell DNA; 3: B95-8 cell DNA; M: DL2000 marker; (**B**) RPA-LFA validation (forward primer labeled with FITC, reverse primer labeled with biotin). a: ddH2O; b: HeLa cell DNA; c: B95-8 cell DNA; (**C**) RPA-LFA validation (reverse primer labeled with biotin and addition of nfo, nfo buffer, and the probe). 1: ddH_2_O; 2: HeLa cell DNA; 3: B95-8 cell DNA; M: DL2000 marker; (**D**) Addition of different reagents to the common RPA system. 1: ddH_2_O; 2: HeLa cell DNA; 3: B95-8 cell DNA; 4: B95-8 cell DNA with probe added; 5: B95-8 cell DNA with nfo added; 6: B95-8 cell DNA with nfo buffer added; 7: B95-8 cell DNA and addition of nfo and nfo buffer; M: DL2000 marker; (**E**) RPA system validation (reverse primer labeled with biotin and nfo and probe were added). 1: ddH_2_O; 2: HeLa cell DNA; 3: B95-8 cell DNA; (**F**) RPA-LFA system validation (reverse primer labeled with biotin and nfo and probe were added). (**G**) RPA system validation (two-step method). 1: ddH_2_O; 2: HeLa cell DNA; 3: B95-8 cell DNA; M: DL2000 marker; (**H**) RPA-LFA system validation (two-step method). a: ddH_2_O; b: HeLa cell DNA; c: B95-8 cell DNA. Note: The upper line of the lateral flow dipstick is the quality control line, and the lower line is the detection line, i.e., the presence of only the quality control line indicates a negative result for EBV, and the presence of both lines indicates a positive result.

**Figure 3 viruses-16-00106-f003:**
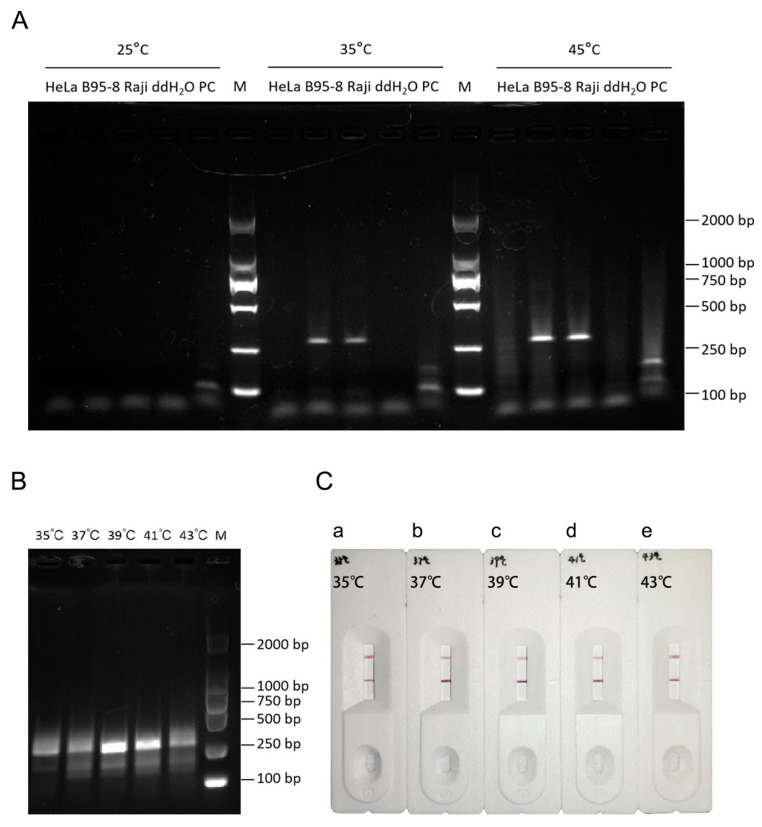
Determination of the RPA-LFA reaction temperature. (**A**) Determination of the optimum RPA temperature for different cell types; M: DL2000 marker; (**B**) Determination of the optimal RPA temperature for B95-8 cell DNA; M: DL2000 marker. (**C**) RPA-LFA validation (two-step method). a: Amplification at 35 °C; b: 37 °C; c: 39 °C; d: 41 °C; and e: 43 °C.

**Figure 4 viruses-16-00106-f004:**
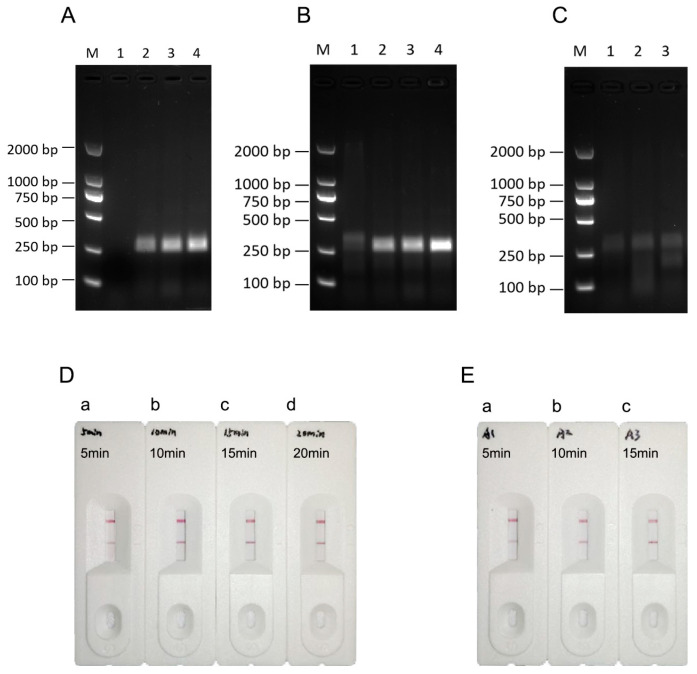
Determination of the RPA-LFA reaction time. (**A**) RPA-conjugated agarose gel electrophoresis to assess amplification using different reaction times. 1: 5 min; 2: 10 min; 3: 15 min; 4: 20 min; M: DL2000 marker; (**B**) RPA-conjugated agarose gel electrophoresis to assess amplification using different reaction times (first step). 1: 5 min; 2: 10 min; 3: 15 min; 4: 20 min; M: DL2000 marker; (**C**) RPA-conjugated agarose gel electrophoresis to assess amplification using different reaction times (second step). 1: 5 min; 2: 10 min; 3: 15 min; M: DL2000 marker; (**D**) RPA-conjugated lateral flow chromatography strips used to assess assays with different reaction times (first step). a: 5 min; b: 10 min; c: 15 min; d: 20 min; (**E**) RPA combined with lateral flow chromatography strips used to assess results for assays with different reaction times (second step). a: 5 min; b: 10 min; c: 15 min.

**Figure 5 viruses-16-00106-f005:**
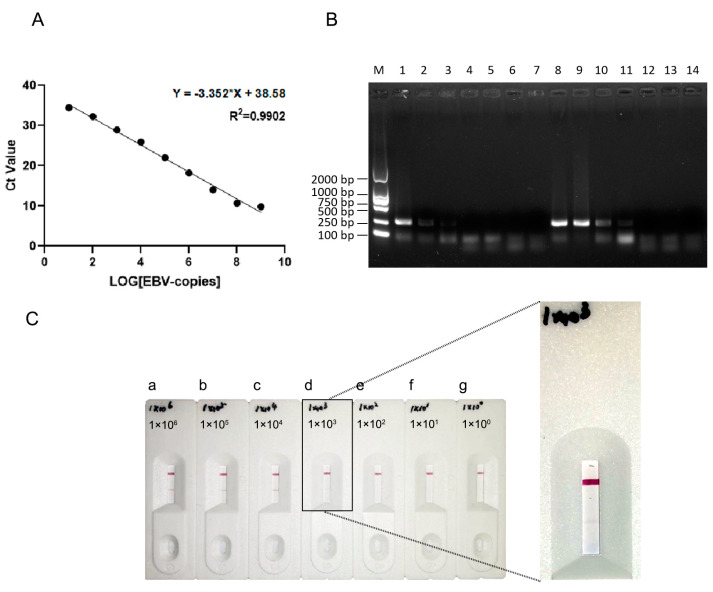
Comparison of the sensitivity of the PCR and RPA-LFA methods for EBV detection. (**A**) EBV standard curve. (**B**) Sensitivity of EBV detection by the PCR method and RPA method. M: DL2000 marker; 1: 1 × 10^6^ copies/μL; 2: 1 × 10^5^ copies/μL; 3: 1 × 10^4^ copies/μL; 4: 1 × 10^3^ copies/μL; 5: 1 × 10^2^ copies/μL; 6: 1 × 10^1^ copies/μL; 7: 1 × 10^0^ copies/μL; M: DL2000 marker. (**C**) Sensitivity of EBV detection by the RPA-LFA method. a: 1 × 10^6^ copies/μL; b: 1 × 10^5^ copies/μL; c: 1 × 10^4^ copies/μL; d: 1 × 10^3^ copies/μL; e: 1 × 10^2^ copies/μL; f: 1 × 10^1^ copies/μL; g: 1 × 10^0^ copies/μL.

**Figure 6 viruses-16-00106-f006:**
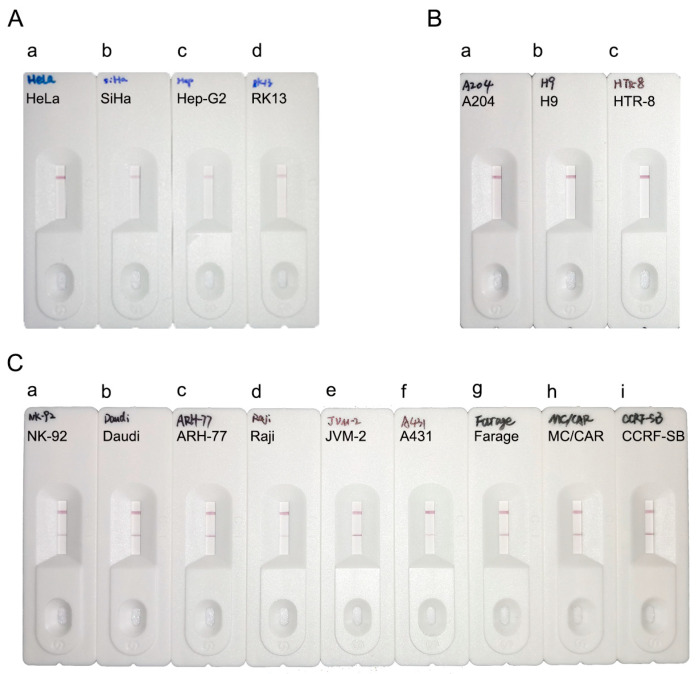
Verification of the specificity of the RPA-LFA system. (**A**) Detection of cells containing other endogenous viruses using the RPA-LFA system. a: HeLa; b: SiHa; c: Hep-G2/2.2.15; d: RK13; (**B**) RPA-LFA detection of EBV-negative cell lines. a: A-204; b: H9; c: HTR-8; (**C**) RPA-LFA detection of EBV-positive cell lines. a: NK-92; b: Daudi; c: ARH-77; d: Raji; e: JVM-2; f: A-431; g: Farage; h: MC/CAR; i: CCRF-SB.

**Figure 7 viruses-16-00106-f007:**
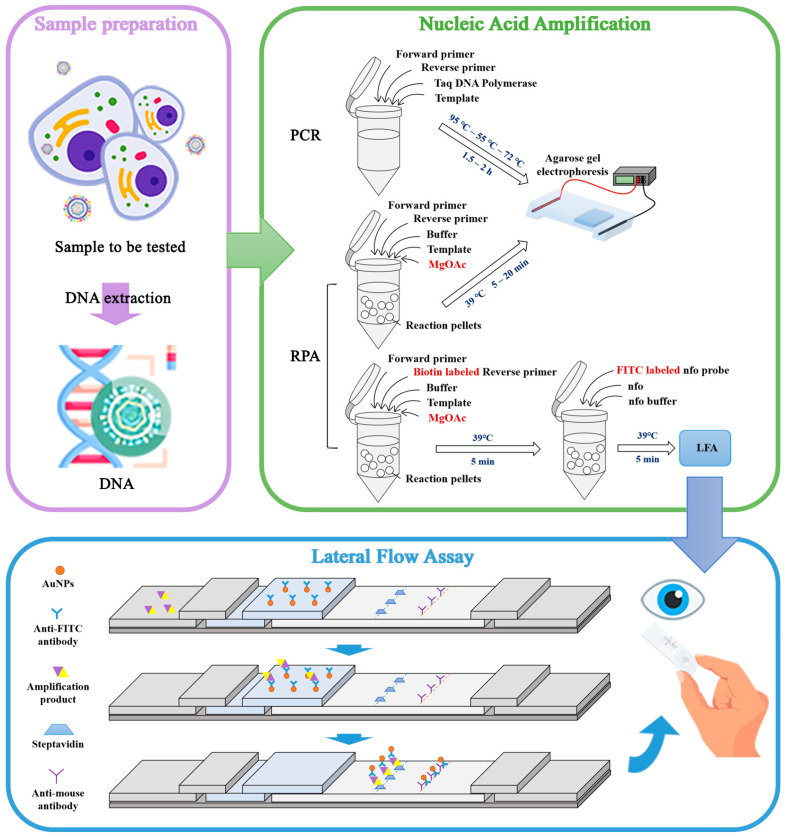
Pattern of EBV detection by the PCR, RPA, and RPA-LFA systems. The three steps, sample preparation, nucleic acid amplification, and the LFA, are shown.

**Table 1 viruses-16-00106-t001:** PCR primer sequences.

Genebank	Nucleotide Positions	Primer	Sequence(5′-3′)
V01555	83,520	PCR-FP1	CTTGGAGACAGGCTTAACCAGACTCA
	−83,784	PCR-RP1	CCATGGCTGCACCGATGAAAGTTAT
	82,507	PCR-FP2	GTGCCTCCTCAAATGTTCCAGAAGT
	−82,737	PCR-RP2	TAAACTGAATCTCCACCTGTGTAACCTCA
	165,806	PCR-FP3	CTACCTGTGCCGCATGAAACTGGGCGAGACCGA
	−165,904	PCR-RP3	CATGTCACAGTAAGGACAGAGAAGTCTGGG
	82,339	PCR-FP4	AGTTAGCATTGGCGTCGG
	−82,497	PCR-RP4	GGAACGGTGATTAGGCACTG
	4683	PCR-FP5	CCTGGTCATCCTTTGCCA
	−4777	PCR-RP5	TGCTTCGTTATAGCCGTAGT

**Table 2 viruses-16-00106-t002:** RPA primers and probe sequences.

Genebank	Primer	Sequence (5′-3′)
V01555	RPA-FP1	[FITC]CTTGGAGACAGGCTTAACCAGACTCA
	RPA-FP2	CTTGGAGACAGGCTTAACCAGACTCA
	RPA-RP	[BIOTIN]CCATGGCTGCACCGATGAAAGTTAT
	RPA-Probe	[FITC]TGCCGGCCCCTCGAGATTCTGACCGGGGACC[THF]CTGGTTGCTCTGTTG[C3-Spacer]

The 5′ end of the reverse primer was labeled with biotin, the 5′ end of the probe was labeled with FITC, and one T base in the middle was replaced with tetrahydrofuran (THF). The 3′ end was attached to the C3 spacer blocking group to prevent DNA strand extension.

## Data Availability

The data that support the findings of this study are available from the corresponding author upon request.

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
