# Peer review of "Development of a Rapid Epstein–Barr Virus Detection System Based on Recombinase Polymerase Amplification and a Lateral Flow Assay"

_viruses, 2024, doi:10.3390/v16010106_

Round 1

Reviewer 1 Report (Previous Reviewer 2)

Comments and Suggestions for Authors

The authors have satisfied my concerns.

Author Response

Thank you for your positive comment.

Reviewer 2 Report (New Reviewer)

Comments and Suggestions for Authors

In the present study, Sun Y. et al. report a rapid Epstein-Barr virus detection system based on recombinase polymerase amplification and a lateral flow assay. In general, the manuscript is written well and the results are appropriately presented. The study is carefully designed.

My comments are listed as follows

1.     I strongly suggest the author provide a Figure to illustrate the region of the detected fragments of the EBV genome.

2.     In addition, as well known that genomic instability accumulates during EBV infection. Have the authors performed any conservation analysis of the EBV genome? Especially for the region detected by the RPA-LFA system proposed in this study.

3.     The RPA-LFA system is only used for detecting EBV-positive or negative cell lines. Have the authors detected any clinical samples using this system?

Author Response

Dear reviewer,

Thank you for the comments and suggestions of the manuscript viruses-2786811. We really appreciate for further improvement of the manuscript. We have carefully read them and make corrections according to your advices. All the changes in the manuscript have been colored gray.

Comments:

  1. I strongly suggest the author provide a Figure to illustrate the region of the detected fragments of the EBV genome.

Many thanks for your suggestion, we have added a figure to illustrate the region of the detected fragments of the EBV genome in Figure 1C.

  1. In addition, as well known that genomic instability accumulates during EBV infection. Have the authors performed any conservation analysis of the EBV genome? Especially for the region detected by the RPA-LFA system proposed in this study.

Thanks a lot for your question, yes, we have performed conservation analysis for the region detected by RPA-LF system. The target fragment is extremely conserved, and many other EBV detection system also used this area to design primers (Ref 15).

  1. The RPA-LFA system is only used for detecting EBV-positive or negative cell lines. Have the authors detected any clinical samples using this system?

Thanks for your suggestion, we haven’t tested any clinical samples, the established RPA-LF method for EBV detection is mainly for the cell lines quality routine check in labs. However, this is a good suggestion, we would like to develop it in clinical practice in the future.

This manuscript is a resubmission of an earlier submission. The following is a list of the peer review reports and author responses from that submission.

Round 1

Reviewer 1 Report

Comments and Suggestions for Authors

The study performed by Sun et al aimed at developing rapid molecular detection systems for cell culture contaminating Epstein-Barr virus. In doing so, the authors focused on developing an isothermal amplification assay (RPA), which was further exploited for dipstick visualization by combining it with lateral flow (LFA) system. While the idea of developing rapid molecular tests to detect contaminating EBV in cell lines will certainly add to the prospective area of application of isothermal RPA, the current version of the manuscript suffers from a number of issues with regard especially to methodological and analytical clarity, which must be addressed.

Major issues:

Since the study is about a molecular assay development, the results regarding the performance of the newly developed assay must be supported by comparative evaluation with an established (e.g., gold standard) and ideally a molecular assay, to determine the sensitivity and specificity measures. Since the study results indicated that purified nucleic acid extracted from the cell culture works best for the proposed assays, the comparison with other established molecular method that uses extracted and purified nucleic acids would be credible. Without this information, the positivity and specificity accounts measured in this study would not contain scientific significance. The authors seemed to have compared the performance only among their own developed PCR, RPA and RPA-LFA assays. Since the authors stated that there are already developed RAA methods (which is basically similar to RPA in principle and operation) for EBV available, the performance of the RPA method should be compared with the published RAA method, which would establish why using the RPA assay developed in this study would be comparable/advantageous for detecting EBV in cell culture to the already developed RAA assay.

The authors have utilized the nfo chemistry of RPA in their development of RPA-LFA system. The RPA-nfo chemistry introduced by Twist-Dx have previously been utilized to standardize different RPA-nfo protocols. To my knowledge, no manufacturer is still producing RPA-nfo reagents. The authors must discuss the reproducibility of the proposed assay in this regard, and detail the production and standardization of any in-house nfo reagents if used.

General suggestion:

The standard of writing, arrangement and formatting of the manuscript, especially the abstract, result and discussion parts, is not acceptable in its current form. The information about study results in the abstract is unclear. Redundancies in wording within and between the main result and discussion sections of the manuscript are noticeable; the result section must be made concise without withholding critical points. Some images (pointed out below) are not clear and must be reproduced. I would recommend the authors find similar articles on RPA/RAA assay development published before to improve the text and organize the sections mentioned above.

Comments:

Line 2: Since this study is about development and not much about implementation, I would suggest changing the title word “Establishment” to “Development”. Also, write the full form of EBV.

Line 14-15: It is not clear what is meant by “sensitivities” in this sentence. Also, “10 times higher” does not make any sense without stating what it refers to. Please clarify.

Line 52-54: Reference is needed for this statement.

Line 77-82: Please provide information with reference to the PCR/real-time PCR assays developed for the detection EBV so far.

Line 85-86: Please provide the citation of the article by Wang et al.

Line 88-89: Please provide the reference of article(s) that performed whole-genome sequencing-based detection of EBV.

Line 94: State the full form of “EBER”.

Line 142 (section 2.3): Please provide more information on the target (length, multicopy or single copy gene fragment, how conserved is the sequence), how were the primers designed and how was the sequencing done.

Line 169: Change “gene” to “genome”.

Line 221: Please provide more information on nfo and whether it was produced or bought (with company name and place).

Line 228 (table 2): Primer names and sequences are not aligned row-wise. Was the second sequence not labeled with FITC?

Line 219-223 (section 2.7.1) and line 236-243 (section 2.7.2): It is very hard to follow why in section 2.7.1 the condition was set at 39 C for 20 min for the two-step method and again in section 2.7.2 the same method was subjected to optimization in different temperatures and time scales. Please clarify and re-write the methodological steps sequentially if needed.

Line 272-274: The authors stated the utility of negative controls, however, this information is absent in the methodology section.

Line 313: nfo and nfo buffer were written multiple times. Please correct or clarify.

Line 330 (Figure 2A and 2E): These images are unacceptable. Band sizes and intensities of the positive bands are not uniform. Please reproduce the figures.

Line 379: The statement is vague as the electrophoresis bands can only indicate qualitative results through visualization (positive or negative or unclear), not quantitative (such as proportional) results.

Line 416 (Figure 5D): This is a critical point for the limit of detection of the LFA system. For the viral load of 10^3 copies, the resulting band is not as clear (i.e., intermediate) as for the viral loads greater than that. Since the results will be decided by the human eye (LFA), this will be prone to incorrect decision-making. I would recommend the authors perform several repetitions of this dilution alongside negative controls to further clarify the outcome.

Line 459-462: How was it made sure that the indicated cell lines were contaminated with the mentioned viruses? What assays were used to detect the viruses in those cell lines? Please include this information in the methodology section.

Line 464-469: Again, there is no comparison with any established molecular/ gold standard method. Without this comparative evaluation, the sensitivity and specificity information could be biased.

Line 486-487: What current EBV detection system are the authors referring to? Please specify with reference.

Line 494: There are no details about the lateral flow dipsticks: whether they were purchased (then include the information in the methodology section with company information), or whether it was produced in-house (then include the production details in the methodology section)

Line 500-502: Again, the statement is not scientifically appropriate since electrophoresis provides qualitative results only.

Line 515-518: The authors stated that they built their own nfo amplification system, but did not provide any details into the production/purchase of the nfo reagents anywhere in the manuscript.

Line 549-551: The authors should explain how is RPA more stable and mature than the RAA method and what advantages it has. Provide reference if necessary.

Line 554-556: The authors should mention what other possibilities have been referred to for rapid binding of the target.

Line 570-574: The authors should discuss the quantification prospect for EBV by not only the qPCR-based methods but also the isothermal assays like RPA and LAMP, since several published works have made attempts to formulate quantification algorithms for isothermal assays and evaluated it for different pathogens such as Dengue virus (https://doi.org/10.1021/acs.analchem.2c02810), Leishmania donovani (https://doi.org/10.3390/diagnostics11111963), Eriwinia amylovora (https://doi.org/10.1002/eng2.12047) to mention a few.

Line 577-579: The authors should mention what other conditions can be explored in the future to improve the detection sensitivity of their RPA-LFA system.

Reviewer 2 Report

Comments and Suggestions for Authors

Establishment of a rapid EBV detection system based on recombinase polymerase amplification and a lateral flow assay by Sun et al focuses on creation of a new detection system for EBV in cell lines.  Overall they report a new RPA-LFA detection system with greater sensitivity than conventional PCR detection methods using selected primer sets.  The manuscript describes the optimization of the system with an end result that is specific for EBV detection in 192 cell lines.

 Current tests to monitor EBV infection largely consist of viral DNA nucleic amplification, in-situ hybridization, and EBV specific antibody tests.  This article is significant in that they develop a more rapid and new lateral flow detection system.  While this new detection system is an exciting new potential improvement for EBV detection it is still a two-step process limited to DNA detection and has only been demonstrated in cell lines and not with clinical samples.  However, advancement of this system may hold value for detection in clinical samples.

Concerns.

English need improvement in the abstract and introduction sections.  Results and discussion sections are largely fine.

Line 58-62.  Please rewrite and rephrase this section.  The authors state that EBV infection induces transformation but this doesn’t occur in healthy individuals.  They also state that EBV is “easily misdiagnosed and underdiagnosed. Therefore, early diagnosis and rational treatment are extremely important.”  Most EBV infections are subclinical and furthermore there are no real specific therapies for EBV associated diseases.

The authors describe primers in table 1.  Please indicate which gene region of EBV these target.  There are primer sequences currently in use for PCR based detection of EBV.  Are these different?  It is also unclear if these primers outperform ones currently in use.  Also the authors don’t determine if their detection method outperforms TaqMan based PCR methods.

The authors analyzing the “cell precipitate.”  Please describe.

The authors use 5 different primers in figure 1 but it is unclear which of these primers is used in figure 2.  Is it primer #1?

Figure 5B and C.  Are these the same exposure times?  In order to really compare the band intensities these should be run on the same gel.

Figure 5D.  The positive band is quite difficult to see.  Is there an explanation for this? For example the band is quite clear in Figure 2HC.  Is much more DNA used here?

Section 3.7 RPA-LFA specificity evaluation.  Were there any known EBV+ cell lines that tested negative in this study?

Comments on the Quality of English Language

The abstract and introduction need to be improved.

Round 2

Reviewer 1 Report

Comments and Suggestions for Authors

The revised manuscript by Sun et al is improved but the data to validate the sensitivity and specificity measures in detecting EBV-positive cell lines (which must be defined by an established molecular method) is still missing. Therefore, the results seem to lack scientific significance, indicating inadequate proof of the effectiveness of the assay in detecting EBV in a random cell line by the proposed method. It is thus suggested that the authors perform additional validation tests and resubmit the manuscript.

Reviewer 2 Report

Comments and Suggestions for Authors

The authors have largely addressed my concerns but I still have issues with Figure 5.  I previously stated that in order to compare the band intensities that Figure 5B and C needed to be run on the same gel.  This was not done but the authors stated that the exposure times were the same.  There are variations in the band intensities of the marker and I don't feel the statement that the RPA detection method is more sensitive than the PCR method is appropriate.  Also they state that the lower detection limit is 1X10^4 but a longer exposure time might reveal bands at 1X10^3.   Also the band in Figure Dd is very faint.  All of this together does not convince me that the RPA and RPA-LFA methods are more sensitive.  Statements concluding that the RPA and RPA-LFA method are more sensitive should be removed or further experimentation needs to be provided.

Round 3

Reviewer 1 Report

Comments and Suggestions for Authors

Dear authors,

Major issue remains regarding the clarification of the definition of the EBV positive and EBV negative samples (in this case, cell lines). To my understanding, the samples were defined by the authors own designed PCR assay but the assessment regarding the efficacy of this PCR assay compared to an established EBV molecular detection method itself is not available in the manuscript. The comparison in positivity between your own designed PCR and your own designed RPA does not justify the effectiveness of your proposed method to detect EBV in a random cell line, and therefore false negative outcomes would be a possibility. A control method (ideally a gold standard, and if not available, an established molecular detection method for EBV published before) is thus necessary to define the samples. If the proposed method is found less sensitive (compared to the result of an established molecular detection method) as per the authors concern, then the limitations must also be discussed in the text. However experimental results are needed to confirm this first. If the difference in sensitivity between an established molecular method and the proposed molecular method is caused by sample types (i.e., clinical vs cell line; which is to me unlikely as you used purified nucleic acid) this should also be discussed in the text after experimental evaluation. Hence I would like to recommend resubmission of the manuscript after inclusion of additional validation results.

Reviewer 2 Report

Comments and Suggestions for Authors

The authors have satisfied my concerns with the updated figure.